# Differences in Starvation-Induced Autophagy Response and miRNA Expression Between Rat Mammary Epithelial and Cancer Cells: Uncovering the Role of miR-218-5p

**DOI:** 10.3390/cancers17152446

**Published:** 2025-07-23

**Authors:** Mateusz Gotowiec, Antoni Smoliński, Katarzyna Marcinkowska, Wiktor Pascal, Paweł Krzysztof Włodarski

**Affiliations:** 1Doctoral School, Medical University of Warsaw, 81 Żwirki i Wigury Street, 02-091 Warsaw, Poland; 2Department of Methodology, Medical University of Warsaw, 1b Banacha Street, 02-091 Warsaw, Poland; s082979@student.wum.edu.pl (A.S.); s088302@student.wum.edu.pl (K.M.); wiktor.paskal@wum.edu.pl (W.P.); 3Department of Histology and Embryology, Medical University of Warsaw, Chałubińskiego 5 Street, 02-004 Warsaw, Poland; pawel.wlodarski@wum.edu.pl

**Keywords:** breast cancer, mammary gland cancer, autophagy, miRNA, translational research

## Abstract

Breast cancer is a complicated disease with varying molecular processes driving its malignant characteristics. Autophagy, the process by which cells break down old and damaged proteins and other substances, allows them to survive stressful conditions, such as nutrient deprivation. This process is regulated by various small, non-coding ribonucleotides, such as miRNAs. In this study, we demonstrate that rat mammary gland cancer cells exhibit distinct nutrient starvation responses compared to healthy mammary epithelial cells. Our results provide insights into the regulatory role of miRNAs in cancer cells’ resistance to starvation and their use of autophagy. We focus specifically on providing a mechanistic description of the role of miR-218-5p in the starvation response.

## 1. Introduction

Breast cancer (BC) is the second leading cause of cancer-related death among women, accounting for over 42,000 deaths each year in the United States alone [1]. Furthermore, it is projected that by 2040, the number of new cases will increase by 40% to over 3 million each year, together with a 50% increase in the number of deaths [2]. Current treatment strategies for BC include surgery, chemotherapy, radiotherapy, endocrine therapy, targeted therapy, and immunotherapy [3]. Although the treatment of early BC is well-established, achieving high survival rates, metastatic BC is still a problem due to its resistance and adaptations. This requires the investigation of alternative therapies, including gene therapy [4]. Within a wide range of gene therapy strategies, microRNAs (miRNAs)—small, non-coding, single-stranded ribonucleotide molecules, which can regulate the expression of many proteins—are of particular interest, due to their ability to induce multifactorial effects [5]. MRX34, a miR-34a replacement agent, is one of the first drugs targeting miRNA expression currently entering clinical trials, showing that further study of microRNAs regulating BC can provide important insights into new potential therapeutic targets [3,6]. Currently, multiple miRNAs are being tested as new treatment targets, with high expectations for the future [7].

Breast cancer, like all types of cancer, can be characterized by its dysregulated metabolism, which leads to high glucose consumption due to Warburg’s effect, and is associated with alternative strategies cells use to generate energy and obtain necessary nutrients [8]. Although one stressor, hypoxia, occurs in over 90% of solid tumors and is considered one of the hallmarks of cancer, it is still disputed whether and how another stressor, nutrient starvation, affects cancer growth [9]. Some researchers postulate that cancer cells significantly differ from healthy cells in terms of stress resistance, due to the hyperactivation of Ras, Tor, PKA, and other proteins, which negatively regulate stress resistance, making cells more susceptible to oxidative stress, DNA damage, and death in the event of excessive starvation [8]. On the other hand, starvation-resistant cells are associated with a poor prognosis in colorectal cancer, showing that an adaptation to nutrient scarcity may be related to further molecular and phenotypic changes of cancer cells, causing increased malignancy [10]. For example, it has been shown that nutrient starvation promotes the expression of genes involved in the epithelial–mesenchymal transition (EMT) in breast cancer cells [11]. Furthermore, nutrient starvation can directly induce migration and invasion of BC cells, which may be mediated via an upregulated glucose metabolism and increased induction of autophagy [12].

Autophagy is a protein degradation process in which damaged proteins and organelles are separated, delivered to lysosomes, and digested, which allows the maintenance of homeostasis in stress situations [13]. Autophagy has been postulated to play an important part in BC development specifically. For instance, it has been shown that the starvation-induced autophagy response differs between ER-positive and ER-negative cell lines, demonstrating that these metabolic adaptations may be associated with other pheno- and genotypic specifications [14]. What is more, it has been proven that BC cells, which have an autophagic induction potential, can adapt to nutrient scarcity faster, selecting cells with specific properties [15]. One mechanism through which BC cells adapt to starvation is through MANF (mesencephalic astrocyte-derived neurotrophic factor), which induces mitophagy (a selective form of autophagy responsible for damaged mitochondria degradation) and enables cells to increase glucose availability [16]. These findings show that autophagy plays a pivotal role in the stress regulation of cancer cells.

The way in which autophagy is regulated includes miRNAs, which can influence a wide range of molecular pathways and commonly act as ‘master regulators’. An example of this regulation can be found in BC. It consists of a functional feedback loop induced in response to nutrient starvation, beginning with a change in phospholipase D expression, and leading to a decrease in autophagy, increased BC cell invasion, and apoptosis inhibition [17]. Furthermore, in lung cancer cells, starvation elicits different miRNA expression changes compared to rapamycin-induced autophagy, showing another example of starvation-specific miRNA dysregulation [18]. Due to their crucial role in the control of cancer cell survival through the regulation of starvation-induced autophagy, further study of potential miRNAs associated with the starvation response in BC cells is necessary.

One such miRNA, which regulates autophagy and is associated with uncertain outcomes in cancer patients, is miR-218-5p, which is known to affect the PI3K/Akt/mTOR signaling pathway [19]. Available studies provide conflicting results on the role of this miRNA in BC progression. Some point towards its important role as a cancer-promoting OncomiR associated with ErbB2 and EGFR expression upregulation [20], Wnt signaling upregulation [21], and collagen type I deposition regulation [22], all associated with bone metastasis. Others present it as an anti-OncomiR, showing that its lower expression in BC tissue samples is associated with lymph node metastasis and worse clinical outcomes [23], and that delivering miR-218-5p in combination with doxorubicin leads to the inhibition of mitophagy and causes BC cell death [24]. Therefore, further research is required to establish the function of miR-218-5p in BC.

In this study, we describe the differences in the starvation-induced autophagy response and miRNA expression changes between different rat mammary gland cancer cells and healthy mammary epithelial cells. We then present a potential mechanism of action of miR-218-5p in the regulation of the starvation response and autophagy. In this study, rat cell lines have been used due to the need for further translational application of the results and due to the lack of similar reported studies in animal cells.

## 2. Materials and Methods

### 2.1. Cells and Cell Culture

All cells used in the study were derived from rats and are listed below: RBA (CRL-1747, ATCC, Manassas, VA, USA) is an in situ adenocarcinoma of a mammary gland cell line; SHZ-88 (305209 CLS Cell Lines Service, Eppelheim, Germany) is another in situ adenocarcinoma of a mammary gland cell line; HH-16.cl.4 (ACC 358, Leibniz-Institut DSMZ, Braunschweig, Germany) is a metastatic adenocarcinoma of a mammary gland cell line; MECs (primary mammary epithelial cells, RA-6035, Cell Biologics, Chicago, IL, USA) were purchased from suppliers and cultured according to suppliers’ instructions. SHZ-88 and HH-16.cl.4 cells were maintained in RPMI 1640 (Gibco, Thermo Fisher Scientific, Waltham, MA, USA) supplemented with 10% fetal bovine serum (Sigma-Aldrich, Burlington, MA, USA) and 1% antibiotic/antimycotic mix (Gibco, Thermo Fisher Scientific, Waltham, MA, USA). RBA cells were maintained in VLE-DMEM medium (Bio&SELL, Feucht, Germany) supplemented with 10% fetal bovine serum and 1% antibiotic/antimycotic mix. MECs were maintained in Epithelial Cell Medium (M6621PF, Cell Biologics, Chicago, IL, USA), which contained 0.1% epithelial growth factor, 0.1% hydrocortisone, 1% antibiotic/antimycotic mix, and 2% fetal bovine serum. Cells were cultured in a 37 °C and 5% CO_2_ cell culture incubator, In-vitro Cell ES. NU-5820E (NuAire, Kewaunee Scientific Corporation, Statesville, NC, USA).

Additionally, for some of the conditions, Earle’s Balanced Salt Solution (EBSS, Sartorius, Göttingen, Germany) was used to induce starvation, while standard full medium (CM)—different for each cell type—was used as control. The standard starvation procedure consisted of 24 h exposure to EBSS following washing of the cells using phosphate-buffered saline (PBS, Sigma-Aldrich, Burlington, MA, USA), if not stated otherwise. Furthermore, for some experiments, 60 μM chloroquine (CQ, chloroquine diphosphate salt, 98%, Thermo Fisher Scientific, Waltham, MA, USA) was added as an autophagosome–lysosome fusion blocker (late autophagy inhibitor) or 500 nM rapamycin (RAPA, Sigma-Aldrich, Burlington, MA, USA) was added to induce autophagy for 24 h.

All cells were tested for the presence of mycoplasma using LookOut^®^ Mycoplasma PCR Detection Kit (Sigma-Aldrich, Burlington, MA, USA) prior to experimental procedures and were found to be negative (Appendix A).

### 2.2. Proliferation Assays (Colorimetric and Colony Formation Assay)

For colorimetric assay, 5120 cells were seeded onto a 96-well plate at a high density. On the next day, cell medium was changed to either EBSS or CM. Following 24 h treatment, the medium was changed back to CM in all wells and the first measurement (24 h) was performed using Cell Counting Kit 8 (WST-8/CCK-8, ab228554, Abcam, Cambridge, United Kingdom). Next measurements were made on the following days (48–96 h). Measurements were performed in quadruplicates. The experiment was independently repeated three times.

For the colony formation assay, 1000 cells were seeded in 6-well plates in cell medium (CM). On the next day, cells were washed twice with PBS and the medium was changed to EBSS, EBSS + CQ, CM, or CM + CQ for 24 h. Afterward, cells were washed twice with PBS and the medium was changed back to CM. Cells were incubated for 7 days with medium changed every 3 days. Then, they were stained with 0.1% Crystal Violet (Warchem, Zakręt, Poland). The photos were analyzed using ImageJ 1.54p 17 (National Institutes of Health, Bethesda, MD, USA) and the ColonyArea plugin [25]. Measurements were performed in triplicate. The experiment was independently repeated three times.

### 2.3. Migration and Invasion Assays

For the migration assay, 50,000 cells were seeded into a single 8 µm 24-well plate insert chamber (Nunc, Thermo Fisher Scientific, Waltham, MA, USA) with 300 µL of CM, EBSS, or EBSS + CQ. The lower chamber was filled with an appropriate CM. After 24 h, the insert was removed, cells were washed twice with PBS, the insert was cleaned using cotton swabs, and stained. Pictures of three random fields (20×) were taken and the number of migrated cells was calculated. Two wells were used for each treatment, and the experiment was independently repeated two times.

For the invasion assay using QCM ECMatrix Cell Invasion Assay (Merck, Darmstadt, Germany), cells were seeded with CM, EBSS, or EBSS + CQ onto the previously hydrated inserts filled with extracellular matrix to simulate the invasion of cancer cells through the basement membrane. CM was added to the lower chamber and cells were incubated for 24 h. Afterward, inserts were washed twice with PBS and cleaned using cotton swabs and stained. Pictures of three random fields (20×) were taken and the number of invasive cells was calculated. A single well was used for each treatment, and the experiment was undertaken once.

### 2.4. Autophagy Assessment

Autophagic flux was measured using an Autophagy Assay Kit (ab139484, Abcam, Cambridge, United Kingdom); 5000 cells were seeded in CM onto one of eight wells in an 8-well cell culture slide (SPL Life Sciences, Pocheon, Republic of Korea). On the following days, medium was changed to EBSS (48 h, 24 h, and 4 h prior to measurement), and to some wells, chloroquine and rapamycin were added. After treatment, cells were incubated with Dual Detection Reagent consisting of Green Detection Reagent—a cationic amphiphilic tracer which specifically stains pre-autophagosomes, autophagosomes, and autophagolysosomes, but not lysosomes—and Hoechst 33,342 Nuclear Stain provided in the kit Following incubation, pictures of three random fields (20×) were taken using a fluorescent microscope, Nikon ECLIPSE Ti-TimeLapse (Nishi-Ōi, Shinagawa, Tokyo, Japan). The experiment was independently repeated two times.

Additionally, Autophagy Essentials Antibody Kit (Proteintech, Planegg-Martinsried, Germany), from which mouse monoclonal antibody against Beclin-1 (66665-1-Ig) was used, assessed the autophagic flux following 24 h, 4 h EBSS starvation, 24 h rapamycin treatment, 24 h chloroquine treatment, and with no treatment. Rabbit polyclonal antibody against beta-actin (NB600-503, Novus Biologicals, Biotechne, Minneapolis, MN, USA) was used as a loading control; 30 ug of sample was loaded per well. The experiment was independently repeated two times. The relative expression was calculated using ImageJ Gel Tool.

### 2.5. Prediction of miRNAs Associated with Starvation Response

The STRING database [26] was used to determine protein–protein interactions associated with autophagy and the starvation response. Genes associated with Autophagy (GO Process) with a high interaction score (>0.700) and involved in the response to starvation were selected. The Mir-target-Link database [27] was used to search for known miRNAs associated with the aforementioned proteins. miRNAs with the most interactions (the closest circle) were chosen for analysis.

### 2.6. miRNA Expression Analysis

Cells were seeded and cultured until they reached 70–80% confluence. Afterward, they were washed with PBS twice and treated with either CM or EBSS for 24 h. Afterward, cells were trypsinized and collected for RNA isolation using the miRNeasy Kit (Qiagen, Hilden, Germany). The quality of RNA was assessed using a spectrophotometer (NanoDrop 2000/c, Thermo Fisher Scientific, Waltham, MA, USA) and only samples with A260/A280 > 1.80 were used for analysis. First strand synthesis of 500 ng of each sample was performed using Mir-X First Strand Synthesis (Takara, Kusatsu, Shiga, Japan) according to the manufacturer’s protocol. Quantitative polymerase chain reaction included the selected 27 miRNAs and an endogenous control U6 snRNA or, for mRNA studies, an endogenous control, glyceraldehyde 3-phosphate dehydrogenase (GAPDH) (for list of 5′ primers, see Appendix A). SYBR™ Green PCR Master Mix (Thermo Fisher Scientific, Waltham, MA, USA) along with 3′ RQ primer (Takara, Kusatsu, Shiga, Japan) was used for the reactions, which were performed in 96-well plates in a total volume of 10 μL. Each sample was analyzed in duplicate or triplicate. Reactions were run on Viia 7 (Life Technologies, Thermo Fisher Scientific, Waltham, MA, USA). ΔCt was calculated for each target relative to U6 snRNA. The experiment was repeated independently three times, and ΔΔCt values were calculated relative to healthy cells (CANCER vs. MECs) or to CM-treated cells (EBSS vs. CM). The final results were presented as mean with upper and lower limit based on SEM incorporated into the ΔΔCt calculation or as a fold expression change.

### 2.7. Supplementary miRNA Analysis

Selected dysregulated miRNAs were analyzed against data from the UALCAN database [28], which uses The Cancer Genome Atlas (TCGA) dataset. Comparisons included normal vs. cancer expression and survival curve analyses following miRNA expression studies. Furthermore, selected miRNAs were analyzed regarding survival curves using the OncomiR database [29].

### 2.8. miR-Transfection Studies

RBA and HH-16.cl.4 cells were seeded onto a 6-well plate and cultured until they reached 70% confluency. Afterward, they were transfected using RNAiMAX lipofectamine (Thermo Fisher Scientific, Waltham, MA, USA) and different mirVana (Thermo Fisher Scientific, Waltham, MA, USA) miRNA mimics and inhibitors including mimic negative control (mimic scramble), inhibitor negative control (inhibitor, scramble), hsa-mir-218-5p mimic (MC10328), and hsa-mir-218-5p inhibitor (MH10328) according to the manufacturer’s protocol. After 24 h of transfection, the medium was changed to a fresh one and cells were incubated for another 24 h. Afterward, RNA was isolated using the same method as previously mentioned and first strand synthesis was conducted as mentioned before. The qPCR reaction setup was similar to the aforementioned one, and ΔCt was calculated for each target relative to U6 snRNA or relative to GAPDH (for mRNA studies). The experiment was repeated independently two times. The final results were presented as fold expression change using the ΔΔCt method.

For proliferation studies, a similar approach was used, where 1280 RBA or 2560 HH-16.cl.4 cells were seeded per well in a 96-well plate, and after 24 h, they were transfected using the same method. Twenty-four hours after transfection, cells were washed twice with PBS, and medium was changed to either EBSS or CM, and after another 24 h, i.e., 48 h post-transfection, cells were washed again twice with PBS, and medium was changed back to CM in all wells. Afterward, the first absorbance measurement was performed according to the aforementioned method, and it was marked as “day 1” with the following days marked accordingly. The experiment was conducted in duplicates and was independently repeated three times.

For migration studies, a similar approach was used, where 30,000 RBA or HH-16.cl.4 cells were seeded per well in a 24-well plate, and after 24 h, they were transfected using the same methods. Twenty-four hours after transfection, cells were collected and seeded into a single 8 µm 24-well plate insert chamber (Nunc, Thermo Fisher Scientific, Waltham, MA, USA) with 300 µL of CM or EBSS. The lower chamber was filled with an appropriate CM. After 24 h, the insert was removed, cells were washed twice with PBS, the insert was cleaned using cotton swabs, and stained. Pictures of three random fields (20×) were taken and the number of migrated cells was calculated. A single well was used for each treatment, and the experiment was repeated independently three times.

For autophagy studies, a similar approach was used. Five thousand cells were seeded in CM onto one of six wells in an 8-well cell culture slide (SPL Life Sciences, Pocheon, Republic of Korea). On the following day, cells were transfected with the aforementioned oligonucleotides and incubated for another 24 h. Afterward, the medium was changed to EBSS or CM and cells were incubated for another 24 h. Afterward, the Autophagy Assay Kit was used to determine the autophagic flux according to the previously mentioned method. Pictures of three random fields (20×) were taken using the fluorescent microscope. The experiment was independently repeated two times.

### 2.9. miRNA Target Prediction

A group of miR-218-5p molecular targets was determined using three databases: miRTarBase [30], Target Scan [31], and miRDB [32]. For each of the databases, we considered different inclusion criteria: miRTarBase: relationship between miR-218-5p and molecular target verified through >1 laboratory method; Target Scan: first 200 genes with best score; miRDB: Target Score ≥ 75. The gathered targets were then intersected with Human autophagy database genes (https://www.autophagy.lu/, accessed on 15 June 2025). The group of intersecting genes was then used for gene expression studies. The expression of selected genes was estimated using the previously mentioned method.

### 2.10. Statistical Analysis

Statistical analyses were explicitly adjusted to each studied outcome and typically based on tests recommended by GraphPad Prism 10.3.1 (GraphPad Software, Boston, MA, USA) with varying statistical significance threshold. The tests used, including unpaired *t*-test for two-group comparison, one-way ANOVA adjusted with Tukey’s multiple comparisons test for more than two-group comparisons within a single treatment factor, two-way ANOVA adjusted with Tukey’s or Dunnett’s multiple comparisons test for more than two-group comparisons with more than one treatment factor, were performed in GraphPad Prism 10.3.1. The number of biological and technical replicates was provided in the description of each method used and in figure captions.

## 3. Results

### 3.1. Starvation Limits the Proliferation of Healthy and Cancer Cells and Starvation Resistance Is Dependent on Autophagy

To better understand the effects of starvation on both healthy mammary epithelial and cancer cells, we conducted a series of experiments focusing on proliferation. Firstly, we aimed to select starvation-resistant subpopulations from both normal and cancer cells, similarly to Li et al. [15], however, we were unsuccessful due to a lack of observable changes in the cells’ phenotype. Therefore, we used native cancer and normal cells to assess the effects of short-term nutrient starvation. Our results show that 24 h EBSS starvation reduces proliferation of both breast cancer cells and, to some extent, healthy cells compared to normal cell medium treatment. However, it does not lead to massive cell death and a proliferation halt, indicating a potential nutrient resistance mechanism, especially in the case of cancer cells (Figure 1A). As a result, we wanted to explore whether autophagy is involved in the cell’s starvation resistance mechanism using chloroquine, which is a late autophagy blocker. We showed that autophagy inhibition using chloroquine decreases the proliferation of cancer cells in starvation, but not in normal conditions, showing the dependence of cell survival during starvation on autophagy (Figure 1B,C).

### 3.2. Starvation Promotes the Invasion and Migration of Mammary Gland Cancer Cells and Both Processes Are Dependent on Autophagy

We further studied the effects of starvation on the cells’ migration and invasion potential. Our results show that 24 h EBSS starvation promotes both the migration and invasion of cancer cells, but not of healthy cells (Figure 2A,B). Furthermore, autophagy inhibition using chloroquine reduced such starvation-promoted migration and invasion.

### 3.3. Autophagy Induction Levels Differ Between Cancer Cells and Healthy Cells, and Some Mammary Gland Cancer Cells Have Higher Potential for Autophagy Induction

To further explore the differences in the starvation response between healthy and cancer cells, we focused on autophagy. Our results point towards major differences (*p* ≤ 0.05) in autophagosome formation between some mammary gland cancer cell lines and healthy cells (Figure 3A). Two cancer cell lines (HH-16.cl.4 and RBA) had a higher autophagosome count when exposed to 24 h and 4 h EBSS starvation, as well as to rapamycin and chloroquine in comparison to normal conditions. On the other hand, SHZ-88 cancer cells and healthy cells (MECs) had a moderately increased autophagosome count only when exposed to 4 h starvation or rapamycin (MECs) and chloroquine (both). Furthermore, we measured the changes in an autophagy-associated protein (Beclin-1) levels in the aforementioned conditions (Figure 3B). The results are partially congruent, showing that Beclin-1 was more abundant in 24 h EBSS starved cells as well as chloroquine-treated cells, while rapamycin did not lead to an increase in Beclin-1.

### 3.4. Autophagy-Associated miRNAs Are Mostly Upregulated in Breast Cancer

We wanted to define a miRNA signature associated with both autophagy and nutrient starvation using a modified approach from Spirina et al. [33]. Therefore, we determined a group of most important proteins related to both processes, firstly through identifying all proteins related to autophagy based on Gene Ontology (GO) (biological process: Autophagy) with an interaction score of high confidence (0.700), followed by an analysis by GO (biological process: Response to starvation) with a low false discovery rate. As a result, we identified 45 genes for further analysis (Figure 4A). The selected genes were screened afterward against known miRNAs using the miR-Target-Link database, and a miRNA signature, consisting of miRNAs with most interactions with the preselected genes, was chosen for analysis (Figure 4B). Out of 31 selected miRNAs, 26 have analogs in rats which were analyzed regarding their expression profile and other important features in BC patients based on the OncomiR database. We found that 16 out of 26 miRNAs were found to be upregulated in tumor tissue compared to normal tissue (Figure 4C). Furthermore, many were associated with (here, only the most significant feature, each miRNA counted once) a pathologic T status (7 out of 26), N status (6 out of 26), and M status (9 out of 26), totaling 22 out of 26 having a negative impact on tumor stage and grade.

Additionally, we included miR-10b-5p based on a literature review, as it plays an important role in cancer metastasis and autophagy regulation [34].

Further miRNA expression analysis was undertaken based on the miRNA signature described here, consisting of 27 miRNAs.

### 3.5. Differences in Autophagy-Associated miRNA Basal and Starvation-Induced Expression Between Mammary Gland Cancer Cells and Healthy Cells

We determined the basal expression profile of the preselected miRNA signature in healthy cells compared to rat breast cancer cells. We found statistically significant differences in the miRNA basal expression pattern between different cells (two-way ANOVA, *p* < 0.0001). In total, we found that 7 out of 27 miRNAs had different expression patterns in at least one breast cancer cell line compared to healthy cells (Figure 5A), including miR-9a-5p, miR-19b-3p, miR-218a-5p (rat orthologue of miR-218-5p with same sequence, below described as miR-218-5p), miR-10b-5p, miR-195-5p, miR-26b-5p, and miR-497-5p. Later, we analyzed whether similar expression patterns occur in BC patients in The Cancer Genome Atlas (TCGA) using the UALCAN database and found that six out of seven miRNAs had similar expression patterns. The only miRNA with an opposite expression profile in rat cells compared to BC patients was miR-9a-5p.

To determine the functional changes of miRNA signature expression, we exposed the cells to 24 h EBSS starvation and compared the miRNA expression to cell medium (CM)-treated cells. Firstly, we tested whether starvation causes an excessive change in housekeeping gene expression—U6 spliceosomal RNA (U6), glycerinaldehyd-3-phosphat-dehydrogenase (GAPDH), and beta-actin. We found no significant change in U6 expression following starvation compared to non-starved conditions in all cells (Appendix A) and among the three housekeeping genes, U6 had the most stable expression pattern (Appendix A). Therefore, we used U6 to calculate the relative expression of miRNAs. Our results show that many of the miRNAs were dysregulated following starvation (Figure 5B) in at least one cell type, including significantly miR-9a-5p and miR-218-5p (unpaired *t*-test, *p* < 0.05), but also miR-503-5p, miR-15a-5p, miR-375-3p, miR-335, miR-195-5p, and miR-20b-5p (unpaired *t*-test, 0.05 > *p* > 0.10).

We found that miR-218-5p expression increased in all breast cancer cells exposed to starvation (unpaired *t*-test), reaching levels from 1.72 (SHZ-88, *p* = 0.20), through 2.02 (RBA, *p* = 0.01) up to 2.12 (HH-16.cl.4, *p* = 0.07), compared to healthy cells, in which the expression decreased to 0.91 (*p* = 0.83) when exposed to starvation (Figure 5C).

### 3.6. Starvation-Upregulated miRNAs in Cancer Cells Are Associated with a Lower Survival Rate in Breast Cancer Patients

Our results show that some of the miRNAs were upregulated in starved breast cancer cells but not healthy cells, i.e., miR-9a-5p and miR-218-5p, while others were upregulated in healthy cells but not cancer cells: miR-503-5p (Figure 5B). To gain clinical relevance of these results, we analyzed them using the OncomiR TCGA BC patients’ database and found that a high expression of miRNAs upregulated specifically in breast cancer cells, i.e., miR-9a and miR-218-5p, is associated with lower survival rates (Figure 6A), while a high expression of miRNA specifically upregulated in healthy cells, i.e., miR-503-5p, is not associated with such negative outcomes (Figure 6B).

### 3.7. The Regulatory Role of miR-218-5p in Proliferation, Migration, and Autophagy in Basal State and During Starvation

As we determined the clinical importance of starvation-upregulated miRNAs in cancer, we focused on miR-218-5p, as its expression change was the biggest and unified across all cancer cell lines. To further analyze the role of miR-218-5p in the starvation response regulation in cancer cells, we conducted transfection studies using cancer cells which had the highest increase in its expression when exposed to starvation. By using a miR-218-5p inhibitor and mimic, we achieved a significant overexpression (mimic) and knockdown (inhibitor) of expression following transfection (Figure 7A). We used the transfected cells to assess the effects on proliferation, migration, and autophagy in both normal and starvation conditions.

Our results showed mostly statistically insignificant changes in proliferation, apart from a subtle acceleration of growth in miR-218-5p overexpressing HH-16.cl.4 cells exposed to starvation at 72 h after seeding compared to scramble-transfected cells (Figure 7B).

Afterward, we determined the effects of miR-218-5p on migration. Our results show that miR-218-5p overexpression promoted cell migration, especially in starvation, while miR-218-5p knockdown inhibited cell migration in starvation (Figure 7C).

Finally, we measured the effects of miR-218-5p on autophagy. The results show that miR-218-5p knockdown led to an increase in autophagosome count, while miR-218-5p upregulation was associated with a lower autophagosome count (Figure 7D).

### 3.8. The Validation of Potential miR-218-5p Targets

Based on the gathered results, we aimed to determine molecular targets of miR-218-5p. Therefore, we employed a strategy to determine autophagy-relevant miR-218-5p targets using three databases (miRTarBase, Target Scan, miRDB) intersected with autophagy-associated proteins from the Human autophagy database (Appendix A). As a result, we found 14 matching genes: one gene occurring in two databases—SNX18, and thirteen genes occurring in separate databases. Out of them, seven most promising genes were selected (SNX18, HMGB1, Eif2ak3, Tp53inp2, CDKN1B, RICTOR, BIRC5) and their expression determined in terms of basal expression, non-starved and starved conditions, and expression change following miR-218-5p modulation.

We found that there were significant differences in the basal expression of preselected genes between different cell types (two-way ANOVA, *p* = 0.0207) with CDKN1B expression being significantly lower in RBA cells compared to MECs (Tukey’s multiple comparisons test, *p* = 0.0330) (Figure 8A). Furthermore, the expression of some of the genes significantly or almost significantly changed (*t*-test, *p* ≤ 0.10) following starvation, with biggest changes occurring in the expression of SNX18, HMGB1, Tp53inp2, and RICTOR (Figure 8B). We noticed an inversely proportional change in the expression of SNX18 compared to miR-218-5p, as following starvation, its expression increased to 1.73 in MECs (*t*-test, *p* = 0.02) and decreased to 0.91 in SHZ-88 cells (*t*-test, *p* = 0.55) and to 0.52 in RBA cells (*t*-test, *p* = 0.006).

Finally, we compared the expression of preselected miR-218-5p targets following miR-218-5p transfection using mimics and inhibitors. Again, we found that the biggest difference concerned SNX18, with a strong decrease in expression in miR-218-5p mimic-treated cells and an increase in miR-218-5p inhibitor-treated cells (Figure 8C,D), especially in one of the cell lines used. As such, miR-218-5p expression again had an inverse relationship to SNX18 expression. The changes in expression of other miR-218-5p targets were not directionally congruent.

## 4. Discussion

This is the first study which describes the effects of starvation and miRNA expression changes in rat mammary gland cancer cells compared to healthy mammary epithelial cells.

Our results show that 24 h total nutrient starvation using EBSS reduced the proliferation of all cells; however, BC cells were able to survive in starving conditions, except when autophagy was blocked. In our study, we used chloroquine to block autophagosome–lysosome fusion. This molecule is a well-known autophagy inhibitor, yet it has been shown that chloroquine can sensitize BC cells to chemotherapeutics, independently of autophagy [35], and inhibit hepatocarcinoma cells in non-starved conditions via an intracellular accumulation of lipids and an inhibition of energy synthesis [36]. However, in our study, we showed that CQ did not significantly reduce proliferation of cancer cells when not starved, showing that in this case, CQ acted primarily through autophagy inhibition in starved cells. We were unable to study whether a similar adaptation is present in healthy cells, as these cells did not proliferate in the colony formation assay used.

Furthermore, we showed that nutrient starvation promotes both the migration and invasion of BC cells but not healthy cells, and such starvation-induced changes were inhibited by an autophagy blockade. Similar results have been recently presented by Wang et al. [12], who, using MDA-MB-231 and BT549, showed that such a starvation-induced increase in migratory capacities is present in human BC cells.

What is more, as we showed that autophagy contributes to cell survival during nutrient starvation, we focused on assessing the differences in the autophagy response of BC cells and healthy cells. Our results point towards differences between specific cell types, but also generally between healthy and cancer cells when it comes to the autophagic response. A similar study, conducted by Zhu et al. [14], showed that comparable differences between cell types were present in human estrogen receptor (ER)-positive and ER-negative BC cell lines. In our case, since we used rat cells, two cell lines—RBA, an ER-negative line, and HH-16.cl.4, a postulated HER2+ metastatic model [37]—achieved a similar and generally high autophagic flux potential. Such results correspond to those gathered by Li et al. [15], who showed that a human metastatic BC cell line—MDA-MB-231—was especially sensitive to autophagic induction, a characteristic not present in other non-metastatic BC cells and non-carcinoma cells.

Furthermore, based on a bioinformatic approach, we established a starvation/autophagy miRNA signature with most miRNAs being differently expressed in cancer compared to healthy tissue. These results point towards the importance of the starvation response modulation through miRNA by cancer cells and its association with negative predictive factors. It has already been shown that cancer cells adapt to nutrient scarcity, which promotes phenotypic changes—leading to increased migration and EMT [38], and that major regulators of this metabolic reprogramming are miRNAs. It is known that specific miRNAs actively regulate the starvation stress response and are expressed at different levels in healthy and cancer cells [39,40]. We have shown that the basal expression of miR-9a-5p, miR-19b-3p, miR-218-5p, miR-10b-5p, miR-195-5p, miR-26b-5p, and miR-497-5p is different in cancer cells compared to healthy cells.

What is more, we have also shown that starvation induces expression changes of some preselected miRNAs, including miR-9a-5p and miR-218-5p, miR-503-5p, miR-15a-5p, miR-375-3p, miR-335, miR-195-5p, and miR-20b-5p. Based on the results, we created a survival curve, which showed that starvation-upregulated miRNAs in cancer cells were associated with worse survival outcomes for patients, while those upregulated in healthy cells were not. This points towards the importance of adaptive changes in miRNA regulation in cancer cells, which potentially allows them to survive such harsh conditions. Out of all studied miRNAs, the most significant results concerned miR-218-5p.

Most of the studies concerning miR-218-5p present it as an anti-oncomiR, with many tumor suppressor properties, ranging from the regulation of EGFR signaling [41] and EMT [42,43,44], but also mitophagy [24] and autophagy [45]. Recent work by Monti et al. [46] showed that zebra-fish extracts caused an upregulation of miR-218-5p in breast cancer cells, which altered the distribution of phosphoinositides, consequently limiting the cells’ malignant phenotype. In other cancers, such as lung cancer, miR-218-5p was found to directly bind to the 3′ region of EGFR gene, inhibiting their effect on cell growth and angiogenesis [41]. In colorectal cancer, miR-218-5p repressed the expression of FTH1P3, a gene responsible for inducing cell migration and EMT [43]. These results point to a role of miR-218-5p as a stress response regulator, which has already been discussed [47]. On the other hand, it has also been described as a potential oncomiR, affecting Wnt signaling and promoting bone metastases [21,22]. In colorectal cancer, Change et al. [48] have shown that miR-218-5p can be expressed by *Parvimonas micra* in the gut microbiome, leading to downregulation of protein tyrosine phosphatase receptor R (PTPRR) and causing tumor progression, showing that microenvironment-derived miR-218-5p can also affect cancer development. Furthermore, it has been shown to be highly expressed in pediatric patients with sarcomas [49], showing that its function is not one-sided, and potentially may depend on the disease and metabolic status. This may be the reason why the data regarding this miRNA are very conflicting, as it may modulate specific cellular processes in a context-dependent manner.

Our results show that its expression in the basal state is lower in cancer cells compared to healthy cells, which corresponds to data available from tissue samples from the TCGA dataset. However, following starvation, the expression of miR-218-5p increases specifically in cancer cells, but not healthy cells. This drove us to study the influence of miR-218-5p on cancer cells further. The results suggest that miR-218-5p only slightly increases the proliferation of cancer cells in starved conditions, but not in a non-starved state; however, it promotes the migration of cancer cells, especially following starvation. These results partially correspond with those gathered by Chu et al. [20], who showed that an upregulation of miR-218-5p leads to an increase in the proliferation and migration of both MCF-7 and MDA-MB-231 cells. Similar results were also presented by Zhang et al. [50], who showed that miR-218-5p leads to an activation of epithelial–mesenchymal transition via DDX21. We further focused on assessing the role of miR-218-5p in autophagy and determined that miR-218-5p overexpression suppressed autophagy, while miR-218-5p knockdown promoted autophagy. Similar results were observed by Chen et al. in rheumatoid arthritis synovial fibroblasts, where knockdown of miR-218-5p also promoted autophagy [45]. A similar pattern was also reported by Naso et al. [24], who showed that miR-218-5p overexpression inhibits doxorubicin-mediated mitophagy in BC cells, which constitutes an important part of the general autophagic response. Interestingly, although presenting contradictory overall results, Chu et al. [20] showed that miR-218-5p inhibits apoptosis in BC cells, which points towards potential context-dependent outcomes resulting from miR-218-5p-mediated suppression of autophagy.

Finally, we have determined a group of potential molecular targets of miR-218-5p associated with autophagy via a database search and found seven genes. We assessed their expression in different conditions. The results have shown that the basal expression of CDKN1B was different in cancer cells, but for all other genes, we were unable to provide conclusive evidence. We also showed that the expression of some of them changed following starvation, including SNX18, HMGB1, Tp53inp2, and RICTOR. Out of all selected targets, the expression of sorting nexin 18 (SNX18)—a membrane remodeling protein, which takes part in recycling endosomes [51] and acts as a positive regulator of autophagy [52]—was inversely related to the expression of miR-218-5p following starvation in different cells. Furthermore, we studied the expression change of those preselected genes in miR-218-5p overexpression and knockdown models, showing that the target being affected the most, with bidirectional changes, was SNX18. The family of sorting nexins, which take part in cell trafficking, has been shown to have both oncogenic and tumor suppressor functions in different cancers [53,54]. For example, SNX1 was found to regulate the expression of epithelial growth factor receptor (EGFR) [55] and a recent work by Atwell et al. [56] has shown that it can act as a molecular target in breast cancer models. On the other hand, it has been shown that SNX9, which is closely related to SNX18, is expressed at decreasing levels with cancer progression and in general in tumor tissue, pointing towards its tumor suppressive functions in breast cancer [57]. We believe that SNX18 may act as the most important target of miR-218-5p in cancer cell starvation response, and its inhibition may regulate cell trafficking, protecting the cancer cells from autophagy overactivation in nutrient-deprived conditions.

All in all, we believe that miR-218-5p takes part in the starvation-induced autophagy response, regulating the expression of SNX18 among other autophagy-associated proteins, and this pathway is an important protective factor, allowing cancer cells to survive under nutrient scarcity. We postulate that starvation induces the expression of miR-218-5p, which causes a metabolic reprogramming of cells, promoting invasion, migration, and limiting autophagy, protecting it from autophagic cell death (Figure 9). It has already been shown that hypoxia induces miR-218 expression [58], which supports the view that another stressor, like nutrient starvation, can also lead to its expression changes. Although autophagy is necessary for survival in stress conditions, as shown by our results, its overactivation may also promote the maintenance of “noninvasive” properties. miR-218-5p, through an inhibition of autophagy, potentially allows cells to change towards more invasive behavior and, at the same time, protects them from such overactivity and autophagic cell death. This miR-218-5p-mediated autophagy inhibition has already been described in the case of colorectal cancer, where miR-218-5p downregulates YEATS24 expression, leading to cell apoptosis [59]. Here, however, potentially due to the different context of nutrient starvation, its role could be cytoprotective.

However, further research is required to establish the exact function of miR-218-5p in cancer. It is necessary to further determine the miR-218-5p-regulated role of SNX18 in the starvation response and assess its effects on migration, invasion, proliferation, and autophagy regulation. What is more, assessing the differences in miR-218-5p expression in different cancer tissues, including primary tumor, lymph node metastases, and distant metastases, could potentially increase the understanding of the clinical role of this miRNA. A better comprehension of other factors, such as the influence of progesterone and estrogen levels on the changes in starvation response, would also bring a more holistic perception of the process. Finally, it is necessary to assess the potential effects of miR-218-5p in animal models, as they are more relevant for potential clinical translation.

In this study, due to the statistical significance of starvation-induced expression changes of miR-218-5p, no other miRNA and its role in the regulation of the stress response has been studied. Another potential regulator to consider could be miR-9-5p, whose expression also significantly changed only in mammary gland cancer cells. Further research using a wider range of study methods, such as microRNA sequencing, could potentially allow us to find even more important regulators of starvation-response in breast cancer, thus forming new molecular targets.

Although this study provides an in-depth analysis of starvation-associated changes in rat mammary gland cancer cells compared to healthy mammary epithelial cells, it contains numerous limitations. Primarily, the use of rat cell lines instead of human ones, which allows for the direct testing of the hypothesized miR-218-5p antagonists in rat animal models, at the same time, does not allow easy translation to humans. Furthermore, although the rno-miR-218a-5p used here is analogous to hsa-miR-218-5p, it is important to notice that while miRNAs are conserved between species, their functions may significantly differ between humans and rats. What is more, the miR-218-5p effects on starvation require further study, as the effects were relatively modest, especially in one of the studied cell lines. Another important limitation of the study is the lack of Beclin-1 expression comparison to healthy cells, which was not possible due to a limited availability of the primary cells. Another limitation is a single-replicate invasion assay, which, although congruent with the migration results and prior studies, needs to be taken cautiously. Furthermore, the hub miRNA was not identified from the computational analysis, which could potentially show the most important miRNA regulating the starvation response prior to in vitro experiments. Another important limitation of this work is the lack of validation using histopathological studies, which could further show the expression changes in the tumor and its surroundings. Finally, starvation is an active process, and therefore to fully understand the cellular response to nutrient scarcity, it is necessary to employ multiple timepoints or use live-cell imaging solutions, which was not possible in this study. Furthermore, it is necessary to determine whether such adaptations are present solely in cancer cells or also in healthy cells, which would allow us to consider miR-218-5p and pathways associated with it as potential molecular targets.

## 5. Conclusions

This study determined that there are strong differences in the short-term starvation response between healthy rat mammary gland cells and cancer cells in terms of proliferation, survival potential dependent on autophagy, invasion, and migration. Furthermore, we showed that starvation-associated miRNAs are expressed at different levels in cancer cells compared to healthy cells and that 24 h starvation induces miRNA dysregulation. We also showed that starvation-upregulated miRNAs in cancer cells are associated with a lower survival rate of patients. Finally, we determined that miR-218-5p promotes cancer cell growth and migration in a starved state and that it limits autophagy, potentially decreasing the risk of autophagic cell death. This in vitro study has shown the importance of miR-218-5p in the starvation response; however, in vivo studies are required prior to therapeutic targeting.

## Figures and Tables

**Figure 1 cancers-17-02446-f001:**
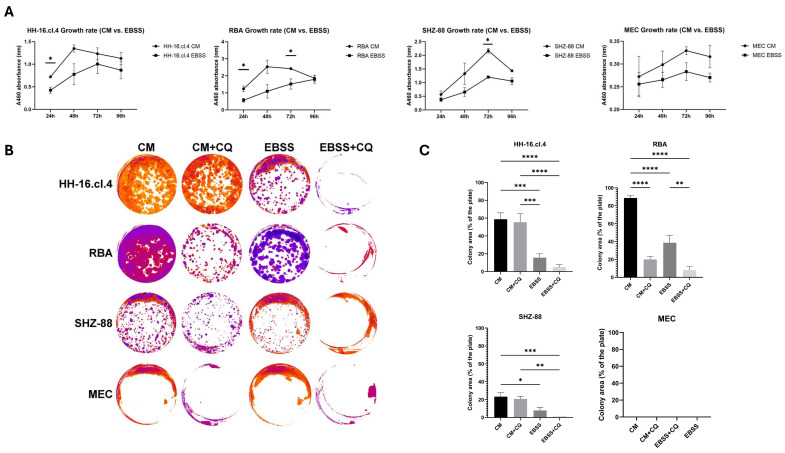
Starvation reduces the proliferation of both healthy and cancer cells, but cancer cells can proliferate following starvation, and their starvation resistance is dependent on autophagy. (**A**) CCK-8 Assay. Three independent experiments with technical quadruplicates. Unpaired *t*-test. *—*p* ≤ 0.05. (**B**) Graphical representation of representative colony formation assay wells with different treatments, including normal conditions (CM), with chloroquine (CM + CQ), starvation (EBSS), and starvation with chloroquine (EBSS + CQ). (**C**) Colony formation assay. Three independent experiments with technical duplicates or triplicates. One-way ANOVA adjusted with Tukey’s multiple comparisons test. *—*p* ≤ 0.05, **—*p* ≤ 0.01, ***—*p* ≤ 0.001, ****—*p* ≤ 0.0001.

**Figure 2 cancers-17-02446-f002:**
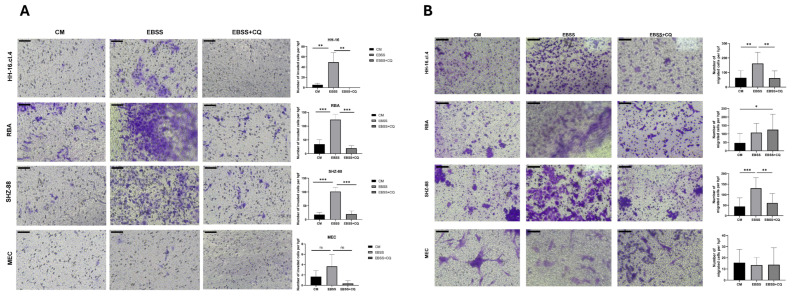
Starvation induces the invasion and migration of cancer cells, which are dependent on autophagy. (**A**) Invasion Assay. One experiment with a technical uniplicate and three representative pictures (20×) taken for each treatment. Standard deviation derived from independent representative pictures. One-way ANOVA adjusted with Tukey’s multiple comparisons test. ns— *p* ≥ 0.05, **—*p* ≤ 0.01, ***—*p* ≤ 0.001. The scale bar represents 100 μm. (**B**) Migration Assay. Two independent experiments with technical duplicates each and two or three representative pictures (20×) taken for each technical replicate. One-way ANOVA adjusted with Tukey’s multiple comparisons test. *—*p* ≤ 0.05, **—*p* ≤ 0.01, ***—*p* ≤ 0.001. The scale bar represents 100 μm.

**Figure 3 cancers-17-02446-f003:**
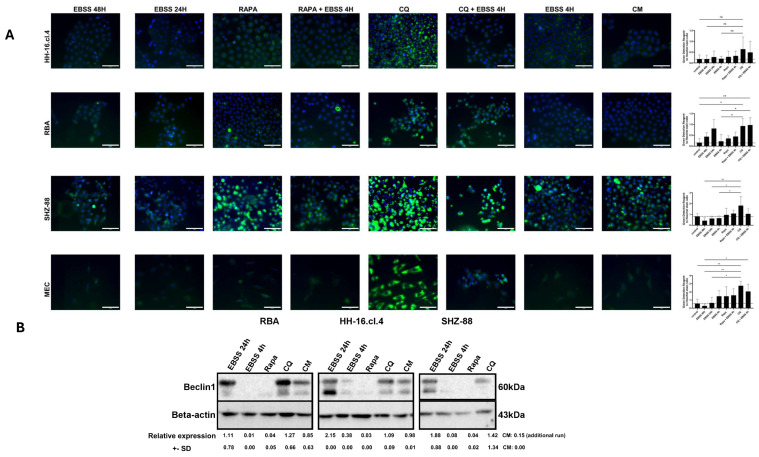
Autophagy is induced in cells exposed to chloroquine, rapamycin, and starvation; however, there are major differences in autophagic flux between healthy and cancer cells. (**A**) Autophagy Assay Kit. Two independent experiments with technical uniplicate and two representative pictures (20×) for each treatment. The scale bar represents 100 μm. One-way ANOVA adjusted with Tukey’s multiple comparisons test. ns—*p* ≥ 0.05, *—*p* ≤ 0.05, **—*p* ≤ 0.01. (**B**) Beclin-1 expression relative to beta-actin. Two independent experiments with technical uniplicate (raw data available from Appendix A). One-way ANOVA adjusted with Tukey’s multiple comparisons test. Significant results (*p* ≤ 0.05) only for RBA: EBSS 24 h vs. all other treatments, EBSS 4 h vs. CQ and CM, RAPA vs. CQ and CM. Data not available for MECs due to low cell availability.

**Figure 4 cancers-17-02446-f004:**
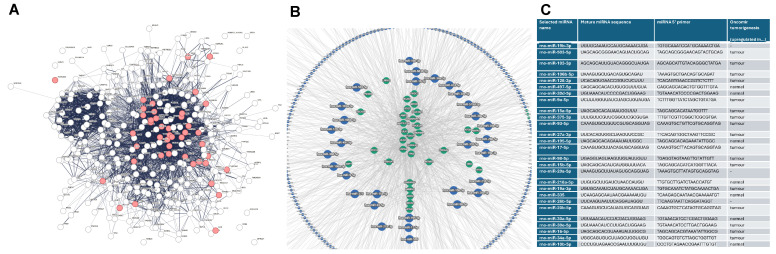
Autophagy-associated miRNAs are dysregulated in breast cancer. (**A**) Protein–protein interaction (PPI) analysis of autophagy and response to starvation. Selected proteins are marked red. (**B**) Mir-Target-Link miRNAs screening against all of the 45 selected genes from PPI analysis. miRNAs forming the inner circle were selected for further analysis. (**C**) List of selected miRNAs-associated with starvation-induced autophagy and their expression pattern determined using the OncomiR database.

**Figure 5 cancers-17-02446-f005:**
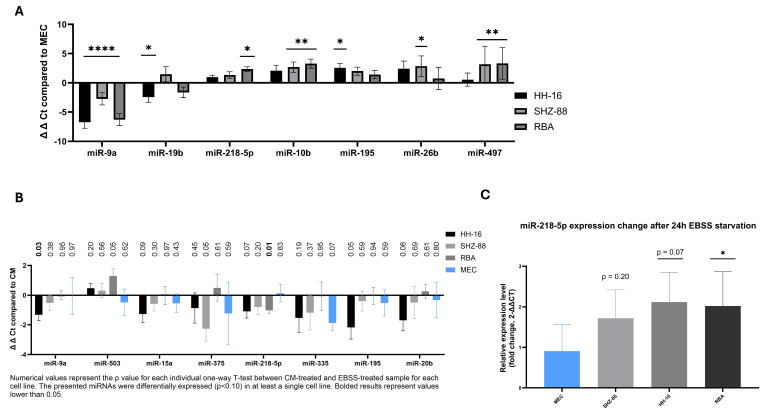
Basal expression of autophagy-associated miRNAs is different in mammary gland cancer cells compared to healthy cells, and starvation induces the expression of miR-218-5p specifically in cancer cells and not healthy cells. (**A**) Relative expression of significantly differently expressed miRNAs in BC cells compared to healthy cells in basal state (CM) presented as ∆∆Ct values. Three independent experiments with technical triplicates. Two-way ANOVA adjusted with Dunnett’s multiple comparisons test was used to determine significant differences in expression of each miRNA in BC cell line compared to MECs. *—*p* ≤ 0.05, **—*p* ≤ 0.01, ****—*p* ≤ 0.0001. (**B**) Relative expression of significantly differently expressed miRNAs in BC cells compared to healthy cells following starvation (CM vs. EBSS) presented as ∆∆Ct values. Three independent experiments with technical triplicates. Unpaired *t*-test for two-group comparison (CM vs. EBSS) was used. Here, presented are all miRNAs with statistically meaningful changes (*p* ≤ 0.10). A *p* ≤ 0.05 was considered to indicate a statistically significant result with such results bolded. (**C**) Relative miR-218-5p expression changes following starvation in mammary gland cancer cells and healthy cells. *—*p* ≤ 0.05. Selected from (**B**).

**Figure 6 cancers-17-02446-f006:**
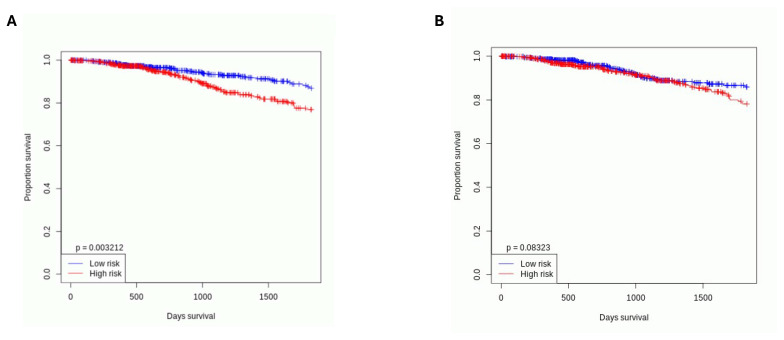
Starvation-upregulated miRNAs in cancer cells (including miR-218-5p) are associated with lower survival rates of BC patients, while those upregulated in healthy cells are not. (**A**) High expression of hsa-miR-9-5p and hsa-miR-218-5p (high risk) is associated with a lower survival rate (*p*= 0.003212) based on univariate Cox analysis. (**B**) High expression of hsa-mir-503-5p (high risk) is not associated with a lower survival rate (*p* = 0.08323) based on univariate Cox analysis.

**Figure 7 cancers-17-02446-f007:**
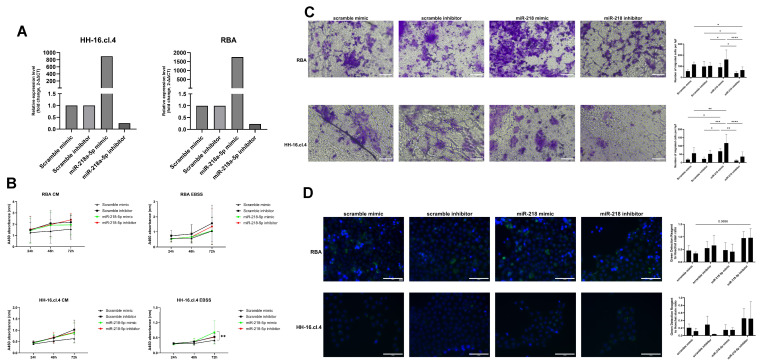
Effects of miR-218-5p on proliferation, migration, and autophagy of cancer cells. (**A**) Change in miR-218-5p expression measured at 48 h post-transfection using 2^−ΔΔCt^. Results presented as averages of two independent experiments with technical triplicates or duplicates. (**B**) Proliferation assay of CM- or EBSS-treated miR-transfected breast cancer cells. Three independent experiments with technical duplicates. Two-way ANOVA with Tukey’s multiple comparisons test. **—*p* ≤ 0.01. (**C**) Transwell migration assay of CM- or EBSS-treated miR-transfected breast cancer cells. Three independent experiments with a technical uniplicate and three representative pictures (20×) for each technical replicate. Pictures presented are of EBSS-treated cells. Two-way ANOVA with Tukey’s multiple comparisons test. *—*p* ≤ 0.05, **—*p* ≤ 0.01, ***—*p* ≤ 0.001, ****—*p* ≤ 0.0001. The scale bar represents 100 μm. (**D**) Autophagy assay of CM- or EBSS-treated miR-transfected breast cancer cells. Two independent experiments with a technical uniplicate and one or two representative pictures (20×) for each treatment. Two-way ANOVA with Tukey’s multiple comparisons test. Pictures presented are of EBSS-treated cells. All comparisons with *p* ≤ 0.10 presented. The scale bar represents 100 μm.

**Figure 8 cancers-17-02446-f008:**
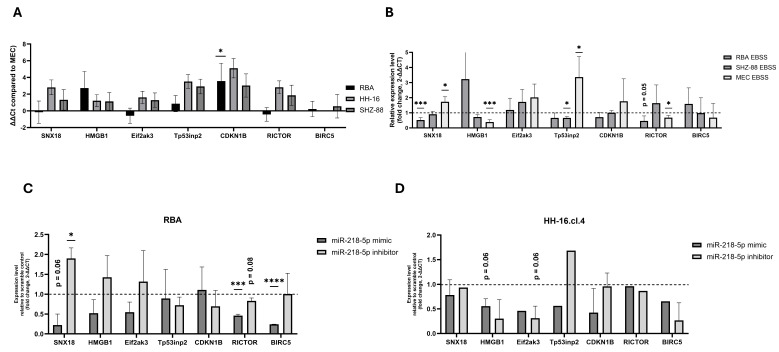
Selected autophagy-associated miR-218-5p target genes expression differences between healthy and mammary gland cancer cells. (**A**) Basal difference in miR-218-5p target genes expression between healthy mammary epithelial cells and cancer cells presented as ∆∆Ct values. Three independent experiments with duplicates and triplicates. Two-way ANOVA adjusted with Tukey’s multiple comparisons test was used to determine significant differences in expression of each miRNA in BC cell line compared to MECs. *—*p* ≤ 0.05. (**B**) Relative expression of miR-218-5p targets in BC cells compared to healthy cells following starvation (CM vs. EBSS) presented as relative expression change of EBSS-treated cells compared to CM-treated cells. Two independent experiments with technical duplicates. Unpaired *t*-test for two-group comparison (CM vs. EBSS) was used; *p*-values given for results with statistically meaningful changes (*p* ≤ 0.10). (**C**,**D**) Change in miR-218-5p target genes expression measured at 48 h post-transfection using 2^−ΔΔCt^ in RBA cells (**C**) and HH-16.cl.4 cells (**D**). Results presented as averages of two independent experiments with technical triplicates. Unpaired *t*-test for two-group comparison (scramble vs. miR-218-5p group each) was used; *p*-values given for results with statistically meaningful changes (*p* ≤ 0.10). *—*p* ≤ 0.05, ***—*p* ≤ 0.001, ****—*p* ≤0.0001.

**Figure 9 cancers-17-02446-f009:**
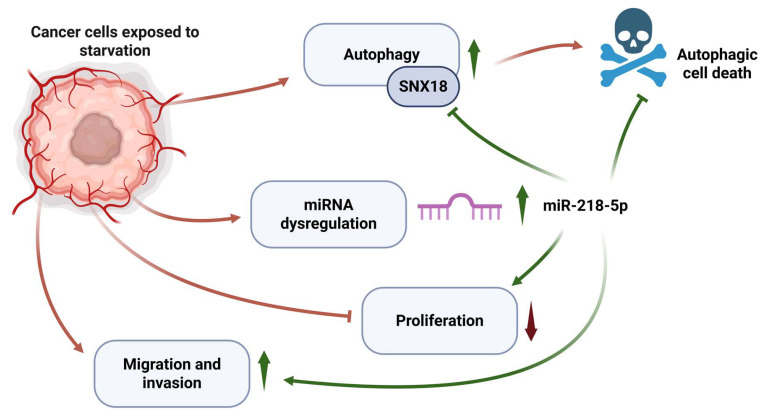
Proposed mechanism of action of miR-218-5p during starvation. Created in BioRender. Pascal, W. (2025) https://BioRender.com/shjzcfu.

## Data Availability

All data are available at request from the corresponding author.

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
