# Peer review of "Differences in Starvation-Induced Autophagy Response and miRNA Expression Between Rat Mammary Epithelial and Cancer Cells: Uncovering the Role of miR-218-5p"

_cancers, 2025, doi:10.3390/cancers17152446_

Round 1
Reviewer 1 Report
Comments and Suggestions for Authors
The authors attempt to demonstrate how miRNA expression induces autophagy in starved breast cancer cells. However, they have not specified the conditions referred to as "starvation." This should be clearly described in the Materials and Methods section. Additionally, the potential influence of estrogen and progesterone levels on autophagy and miRNA activity should be considered and ideally addressed through an additional experiment. Furthermore, the authors are encouraged to explore and present the regulatory role of miRNA-218-5p in autophagy-related pathways, such as the PI3K/AKT/mTOR signaling cascade.
It is also recommended that the overall English language throughout the manuscript be improved for clarity and readability.
Comments on the Quality of English Language
It is also recommended that the overall English language throughout the manuscript be improved for clarity and readability.
Reviewer 2 Report
Comments and Suggestions for Authors
The manuscript investigates the differential response to starvation-induced autophagy and miRNA expression between rat mammary epithelial cells and cancer cells, with a focus on miR-218-5p.
Overall, my assessment is that the study is well-structured and addresses an important topic in cancer biology.
However, there are some major weaknesses which need to be addressed.
I provide constructive feedback for the authors:
- The abstract mentions "significant changes" but does not provide specific fold changes or p-values for key results (e.g., miR-218-5p upregulation).
- Ideas about therapeutic implications are speculative without in vivo validation.
- The introduction jumps between autophagy, starvation, and miRNAs without a clear thread linking them to the study’s hypothesis.
- Organize the autophagy/starvation section to focus on breast cancer-specific mechanisms.
- There are some statistical problems. Migration/invasion assays were performed only once (n=1) with technical replicates, undermining reliability.
- No rescue experiments (e.g., miR-218-5p mimic + target gene overexpression).
- Figure 3B lacks MEC data due to "low cell availability," which is unacceptable for a comparative study.
- There’s no negative controls for miR-218-5p transfection (e.g., scrambled miRNA effects on autophagy).
- Claims about miR-218-5p’s role in migration are based on modest effects (e.g., Fig. 7C shows ~1.5-fold changes).
- Repeat Beclin-1 experiments in MEC or explain exclusion more rigorously.
- The proposed model (Fig. 8) is speculative without target gene validation.
- No discussion of contradictory roles of miR-218-5p in other cancers.
- Discuss conflicting literature on miR-218-5p (e.g., oncogenic vs. tumor-suppressive roles).
- There’s no mention of the single-replicate migration assays.
- Address the lack of mechanistic data (e.g., miR-218-5p target genes).
- Clearly mention that migration/invasion data are preliminary due to low replicates.
- The Discussion doesn’t address why some miRNAs (e.g., miR-9a-5p) were not pursued further.
- Edit conclusions to emphasize the need for further validation (e.g., "miR-218-5p may be a candidate for therapeutic targeting pending in vivo studies").
- The scale bars in some figures are not clear and not mentioned what is the size.
- Figure 5: Use consistent formatting for miRNA nomenclature (e.g., miR-218-5p vs. miR-218a-5p).
- Avoid confusing terms like "major differences" without quantification. Please prrovide statistical data when discussing major differences.
Reviewer 3 Report
Comments and Suggestions for Authors
- What is the prevalence of Breast Cancer among your cohort?
- How have you found the endogenous control- U6nRNA for your study?
- Why have you not identified hub miRNA from the PPI network analysis?
- This study has not been validated on clinical samples. any specific reason for this?
Round 2
Reviewer 1 Report
Comments and Suggestions for Authors
The manuscript may now be accepted in the present form.
Reviewer 2 Report
Comments and Suggestions for Authors
The authors have addressed my comments well.